# Beyond Link Prediction: On Pre-Training Knowledge Graph Embeddings

## Abstract

Knowledge graph embeddings (KGE) models provide low-dimensional representations of entities and relations in a knowledge graph (KG). Most prior work focuses on training and evaluating KGE models for the task of link prediction; the question of whether or not KGE models provide useful representations more generally remains largely open. In this work, we explore the suitability of KGE models (i) for more general graph-structure prediction tasks and (ii) for downstream tasks such as entity classification. For (i), we found that commonly trained KGE models often perform poorly at structural tasks other than link prediction. Based on this observation, we propose a more general multi-task training approach, which includes additional self-supervised tasks such as neighborhood prediction or domain prediction. In our experiments, these multi-task KGE models showed significantly better overall performance for structural prediction tasks. For (ii), we investigate whether KGE models provide useful features for a variety of downstream tasks. Here we view KGE models as a form of self-supervised pre-training and study the impact of both model training and model selection on downstream task performance. We found that multi-task pre-training can (but does not always) significantly improve performance and that KGE models can (but do not always) compete with or even outperform task-specific GNNs trained in a supervised fashion. Our work suggests that more research is needed on the relation between pre-training KGE models and their suitability for downstream applications.

## 1 Introduction

Knowledge graph embeddings (KGE) provide low-dimension representations of entities and relations of a knowledge graph (KG). Although a large number of KGE models have been proposed in the literature—see for example the surveys of Nickel et al. (2015),Wang et al. (2017) and Ji et al. (2021)—, most prior work focuses on the task of link prediction, i.e., answering questions such as *(Austin, capitalOf, ?)* by reasoning over an incomplete KB. In addition to link prediction, it is often argued that KGEs can provide representations that capture semantic properties of the entities and, indeed, pre-trained KGE models have been used to inject structured knowledge into language models (He et al., 2020; Zhang et al., 2019), visual models (Baier et al., 2017), recommender systems (El-Kishky et al., 2022; Wang et al., 2018), question answering systems (Ilyas et al., 2022) and other types of downstream models (Wang et al., 2017).

The question of whether pre-trained KGE models provide generally useful representations remains largely open. Likewise, it is not well-understood how choices taken in model training and model selection affect these representations. In this work, we shed light onto these questions from multiple directions.

First, we study the suitability of out-of-the-box KGE models for basic graph-structure prediction tasks beyond link prediction. In particular, we consider the tasks of predicting the relation of a triple as suggested by Chang et al. (2020) (e.g., the relationship between *Austin* and *Texas*), the domain and range of a relation (e.g., whether *Austin* is a capital), as well as entity and relation neighborhood of each entity (e.g., which other entities are related to *Austin*). Perhaps surprisingly, we found that commonly trained KGE models often performed poorly on such tasks, challenging the intuition that KGE models capture graph structure well.

Second, we investigate whether KGE models are suitable pre-trained representations for node-level downstream tasks such as entity classification (e.g., the profession of a person) or regression (e.g., the average rating of a movie). To do so, we conducted an empirical study using 27 downstream tasks on two different KGs. We found that out-of-the-box KGE models often perform decent on these tasks and, in fact, the best KGE models can (but do not always) exceed the performance of recent graph neural networks such as KE-GCN (Yu et al., 2021). However, the KGE models with best downstream task performance were often not the best-performing models for link prediction. For example, we found that the basic TransE model (Bordes et al., 2013) may be superior to KGE models more suited to link prediction such as ComplEx (Trouillon et al., 2016) or RotatE (Sun et al., 2019). This suggests that link prediction performance is not necessarily indicative of downstream task performance.

Both of these findings suggest that the focus on link prediction tasks is too narrow for pre-training KGE models, i.e., to provide generally useful features.. We thus explore whether the performance of KGE models for both graph-structure prediction and downstream tasks can be improved by better pre-training and model selection. Inspired by multi-task approaches in other areas—such as natural language processing (Aribandi et al., 2022; Sanh et al., 2022) or computer vision (Doersch & Zisserman, 2017)—, we included the graph-structure prediction tasks discussed above as additional training objectives and as evaluation measures during model selection. In particular, we propose a multi-task training (MTT) and a multi-task ranking (MTR) approach that both can be used along with an arbitrary KGE model class and without a substantial increase in computational cost. In our experimental study, the resulting multi-task KGE models had significantly better overall performance for graph-structure prediction tasks and often (but not always) also led to better downstream task performance. We also found that downstream task performance could be further improved by using a smaller set of pre-training tasks. The results suggest that the optimal choice of tasks depends on the dataset, KGE model class, and downstream task and may be difficult to determine in practice.

In summary, the contributions of this paper are as follows: (i) We show empirically that commonly trained KGE models fail at basic graph-structure prediction tasks beyond link prediction. (ii) We propose novel multi-task training and ranking approaches that address this shortcoming. (iii) We explore the impact of standard and multi-task training as well as different approaches for model selection on downstream task performance. (iv) We contextualize KGE model performance with results obtained from recent graph neural networks, which—in contrast to KGE models—are trained directly on each downstream task. Although our work makes a step toward improved pre-training of KGE models, it also suggests that more research is needed on the relation between pre-training KGE models and their general suitability for downstream applications.

## 2 PRELIMINARIES AND RELATED WORK

We briefly describe KGE models, training and evaluation methods for link prediction, as well as prior work on other tasks. A more comprehensive discussion can be found in surveys such as (Nickel et al., 2015; Wang et al., 2017; Ji et al., 2021).

**Link prediction.** A *knowledge graph* $\mathcal{G} \subseteq \mathcal{E} \times \mathcal{R} \times \mathcal{E}$ is a collection of *(subject, predicate, object)*-triples over a set $\mathcal{E}$ of entities and a set $\mathcal{R}$ of relations. Triples represent known facts such as *(Austin, capitalOf, Texas)*. In the KGE literature, the *link prediction* task is the task of inferring the subject or object to questions of form *(?, capitalOf, Texas)* and *(Austin, capitalOf, ?)*, respectively.

**KGE models**. KGE models (Sun et al., 2019; Trouillon et al., 2016; Bordes et al., 2013) represent each entity and each relation of a KG with a a low-dimensional embedding, commonly a real or complex vector. KGE models have an associated *scoring function* s : $\mathcal{E} \times \mathcal{R} \times \mathcal{E} \to \mathbb{R}$ that associates each triple with a real-valued score. Intuitively, high scores indicate plausible triples, low scores implausible triples. Commonly, the scoring function depends on the input triple only through the embeddings of its arguments. For example, TransE (Bordes et al., 2013) is a translation-based model with s$(i, k, j) = -\|\boldsymbol{e}_i + \boldsymbol{r}_k - \boldsymbol{e}_j\|$, where $\boldsymbol{e}_i \in \mathbb{R}^d$ and $\boldsymbol{r}_k \in \mathbb{R}^d$ denote entity and relation embeddings of dimensionality $d > 0$, respectively. Scoring functions can be more involved, e.g., based on convolutional neural networks (Dettmers et al., 2018) or transformers (Chen et al., 2021a).

**Standard training**. KGE models are commonly trained on the link prediction task. We only give a high-level description here. For each triple $(s, p, o)$ in the training data $\mathcal{G}_{\text{train}}$, KGE models are trained

such that the score $\text{s}(s, p, o)$ is high (a positive) but, for certain choices of $o' \in \mathcal{E}$ such that $(s, p, o') \notin \mathcal{G}_{\text{train}}$, the score of $\text{s}(s, p, o')$ is low (a negative); similarly for subjects $s' \in \mathcal{E}$ with $(s', p, o) \notin \mathcal{G}_{\text{train}}$. The actual cost function varies across training types (e.g., sampled negatives or all negatives), loss function (e.g, cross entropy), and more generally the choice of hyperparameters; see (Ali et al., 2021; Ruffinelli et al., 2020) for a more detailed discussion and experimental comparison.

**Standard evaluation**. The most commonly used evaluation protocol for KGE models is *entity ranking* (ER), which is also based on link prediction. Given a test triple $(s, p, o) \notin \mathcal{G}_{\text{train}}$, the model is used to answer the link prediction queries $(s, p, ?)$ and $(?, p, o)$. In particular, the scores of all possible answers that do not already occur in the training data are computed. The model is evaluated based on the rank of the test answers $s$ and $o$, respectively. Common metrics are the mean reciprocal rank (MRR) and Hits@K. The reliability of entity ranking in assessing model performance was studied and questioned, e.g., in (Safavi & Koutra, 2020; Tiwari et al., 2021; Zhou et al., 2022; Wang et al., 2019). In contrast, our focus is mostly on other evaluation tasks.

**Other training approaches**. RESCAL (Nickel et al., 2011), one of the earliest KGE models, trained on the *reconstruction task*. Such tasks aim to to construct the entire training data using cost functions such as $\sum_{s,p,o} \|I[(s, p, o) \in \mathcal{G}_{\text{train}}] - \text{s}(s, p, o)\|_2^2$, where $I[\cdot]$ is a 0/1 indicator. A similar approach was explored by Li et al. (2021). We do not consider such methods further because training costs are excessive (at least unless squared error is used) and the empirical performance reported by Li et al. (2021) is generally far behind KGE models trained with link prediction. Chen et al. (2021b) proposed to augment the link prediction task with *relation prediction* during training (but not evaluation). We expand upon this work by considering additional pre-training tasks and by focusing on graph-structure prediction and downstream task performance instead.

**Other evaluation approaches**. In early (and rarely in recent) work, KGE models were evaluated using *triple classification* (Socher et al., 2013; Wang et al., 2014; Lin et al., 2015; Wang et al., 2022). We do not consider this task in this work because performance estimates are typically overly optimistic and misleading unless hard negatives are used (Safavi & Koutra, 2020); such hard negatives are generally not available. Chang et al. (2020) evaluated KGE models on the relation prediction task, which we also consider as one of the evaluation tasks in this work. There is also work on explaining or interpreting KGE models (Meilicke et al., 2018; Allen et al., 2021; Rim et al., 2021), whereas our focus is on studying whether such models provide useful representations in the first place. As mentioned in the introduction, pre-trained KGE models have been used as a components in language models (He et al., 2020; Zhang et al., 2019), visual models (Baier et al., 2017), recommender systems (El-Kishky et al., 2022; Wang et al., 2018), or question answering systems (Ilyas et al., 2022). Likewise, (Pezeshkpour et al., 2018; Jain et al., 2021) evaluated pre-trained KGE models for entity classification or regression tasks, as we do. We expand on this line of work by using a larger set of tasks (graph-structure prediction and more downstream tasks), by proposing improved pre-training methods, and by studying the impact of pre-training on downstream task performance.

## 3 GRAPH-STRUCTURE PREDICTION

In addition to link prediction, we explore the suitability of KGE models for other basic graph-structure prediction tasks. An example and summary is given in Table 1. We describe the form of the *queries* for each task as a triple such as $(s, ?, *)$, where $s$ or $o$ denote input entities, $p$ denotes an input relation, ? denotes the prediction target, and $*$ acts as a wildcard. Using this notation, we consider the following tasks and queries:

- **Link prediction** (LP): Given a relation and a subject, predict the object (denoted $(s, p, ?)$). Likewise, given a relation and an object, predict the subject (denoted $(?, p, o)$).
- **Relation prediction** (REL, Chang et al. (2020); Chen et al. (2021b): Given two entities $s$ and $p$, predict the relation between them (denoted $(s, ?, o)$).
- **Domain prediction** (DOM): Given a relation, predict its domain (denoted $(?, p, *)$) or its range (denoted $(*, p, ?)$).
- **Entity neighborhood prediction** (NBE): Given a subject entity, predict related objects (denoted $(s, *, ?)$). Likewise, given an object, predict related subjects (denoted $(?, *, o)$).
- **Relation neighborhood prediction** (NBR): Given a entity, predict the relations where it occurs as subject (denoted $(s, ?, *)$) and where it occurs as object (denoted $(*, ?, o)$).

| Knowledge graph | Task | Example query | Some answers |
|---|---|---|---|
| *(Dallas, locatedIn, Texas)* | Link (LP) | *(Austin, locatedIn, ?)* | *Texas, USA* |
| *(Texas, locatedIn, USA)* | | *(?, locatedIn, Texas)* | *Austin, Dallas* |
| *(Austin, capitalOf, Texas)* | Relation (REL) | *(Austin, ?, Texas)* | *locatedIn, capitalOf* |
| *(Austin, locatedIn, Texas)* | Domain (DOM) | *(\*, locatedIn, ?)* | *Texas, USA, North A.* |
| *(Arkansas, borders, Texas)* | | *(?, locatedIn, \*)* | *Dallas, Texas, USA* |
| *(USA, locatedIn, North A.)* | Entity neighb. (NBE) | *(Austin, \*, ?)* | *Texas, USA* |
| *(Austin, locatedIn, USA)* | | *(?, \*, Texas)* | *Dallas, Arkansas* |
| | Relation neighb. (NBR) | *(Austin, ?, \*)* | *capitalOf, locatedIn* |
| | | *(\*, ?, Texas)* | *borders, capitalOf* |

Table 1: Graph-structure prediction tasks used for self-supervised pre-training and evaluation along with example queries. Here ? denotes the prediction target and $*$ acts as a wildcard.

Note that we use the wildcard to denote existential quantification. For example, given a ground-truth KG $\mathcal{G}$ and domain prediction query $(?, p, *)$, an entity $s \in \mathcal{E}$ is a correct answer if there exists an entity $o \in \mathcal{E}$ such that $(s, p, o) \in \mathcal{G}$.

We chose this particular set of tasks because they are simple, they capture basic information about the graph structure beyond link prediction, and they only have one prediction target (an entity or a relation). The latter property allows efficient pre-training and evaluation, as discussed below. For this reason, we exclude tasks such as entity-pair prediction (Wang et al., 2019) (denoted $(?, p, ?)$ in our notation) or reconstruction (Nickel et al., 2011) (denoted $(?, ?, ?)$). In our experimental study, we also found that the exclusion of some of the above pre-training tasks (e.g., LP) can further improve downstream task performance. The optimal choice of tasks depends on dataset, KGE model, and downstream task, however. We leave the exploration of task selection as well as on exploring additional pre-training tasks to future work.

**Multi-task ranking (MTR).** To evaluate the performance of KGE models on the graph-structure prediction tasks, we generalize the entity ranking (ER) protocol for link prediction. Intuitively, for each of the nine tasks (REL as well as LP/DOM/NBE/NBR for both subject targets and for object targets), we construct a query from each test triple,[1] obtain a ranking of the prediction targets (entity or relation) that do not already occur in the training data, and then use metrics such as MRR or Hits@K. The final MTR metric is given by the micro-average over all nine tasks.

We now describe how to obtain task-specific rankings. First, for a REL query of form $(s, ?, o)$, we proceed as in (Chang et al., 2020) and rank all $r' \in \mathcal{R}$ such that $(s, r', o) \notin \mathcal{G}_{\text{train}}$ in descending order of their scores $\text{s}(s, r', o)$. For the other tasks, which involve wildcards, it is not immediately clear how to perform prediction using a KGE model. We first discuss scoring and ranking, then filtering of training data. Consider for example the NBR query $(s, ?, *)$, where our goal is to rank relations. The perhaps simplest approach to obtain a relation ranking is to first rank all triples of form $(s, r', o')$, where $r' \in \mathcal{R}$ and $o' \in \mathcal{E}$, and then rank relations by their first appearance (e.g., the relation of the highest-scoring triple is ranked at the top). More generally, we make use of an *extended score function* that accepts wildcards. The approach just described corresponds to using $\text{s}(s, r', *) = \max_{o' \in \mathcal{E}} \text{s}(s, r', o')$, i.e, the score of a relation $r'$ is the score of its most plausible triple. Although other aggregation functions are feasible, we only consider max-aggregation because it does not make any additional assumptions on the scoring function. To filter training data during model evaluation, we remove all relations $r'$ such that $(s, r', o') \in \mathcal{G}_{\text{train}}$ for some $o' \in \mathcal{E}$; i.e., we remove all prediction targets that are already implied by the training data. We proceed similarly for all other tasks involving wildcards. Note that the number of score computations needed to predict entity targets for queries without wildcards is $O(|\mathcal{E}|)$, whereas the one for queries with wildcards is $O(|\mathcal{E}||\mathcal{R}|)$. We discuss below how the latter cost can be reduced to $O(|\mathcal{E}|)$.

**Multi-task training (MTT).** We now generalize standard KGE model training to all of the graph-structure prediction tasks. Our goal is to be able to improve KGE model performance at these tasks, while at the same time keeping training and prediction cost low. We do this by constructing a task-specific cost function for each individual task first; the final cost function is then given as a weighted

---

[1]The nine queries for test triple $(s, p, o)$ are precisely the ones given in the task descriptions.

linear combination of the task-specific costs (and additional regularization terms), where the weights are hyperparameters.

The task-specific cost functions for link prediction and relation prediction are obtained as in standard training (Sec. 2): For each positive triple $(s, p, o) \in \mathcal{G}$, we construct a set of negatives according to the query (i.e., by perturbing the position of the prediction target) and then apply the loss function (e.g., cross entropy). For the other tasks, which involve wildcards, we proceed differently. Instead of performing some form of (costly) score aggregation during training, we "convert" tasks with wildcards into tasks without wildcards. To do so, we make use of three virtual *wildcard entities*—one for subjects ($any_S$), one for relations ($any_R$), and one for objects ($any_O$)—and learn embeddings for these entities. During training, we conceptually replace wildcards by their corresponding wildcard entity and proceed as before. For example, for training triple $(s, p, o)$ and NBR query $(s, ?, *)$, we consider the virtual triple $(s, p, any_O)$ along with query $(s, ?, any_O)$. By doing so, we converted the NBR task into a REL task. We also use the so-obtained wildcard embeddings during prediction time in the same fashion; e.g., we set $s(s, r', *) = s(s, r', any_O)$. Instead of performing score aggregation, the model thus directly learns extended scores.

The advantage of the MTT approach is that (i) the prediction costs remain stable, i.e., the cost of graph-structure prediction or downstream task prediction is unaffected by the number or choice of pre-training tasks, and (ii) the pre-training costs increase only linearly in the number of tasks.

Note that the wildcard embeddings are not used for entity-level downstream tasks. Nevertheless, using wildcard entities during training affects all other entities as well. This is because the embedding of each entity occurs in all graph-structure prediction tasks. The entity embeddings of a good KGE model thus needs to be suitable for all these tasks, not just for link prediction.

# 4 EXPERIMENTAL STUDY

We conducted a large experimental study. Our goals were (i) to assess KGE model performance for graph-structure prediction, (ii) to assess performance of pre-trained KGE models on downstream tasks, (iii) to assess the effect of multi-task training on both graph-structure prediction and downstream task performance, and (iv) to contextualize these results by comparing them to results obtained by recent graph neural networks (GNNs).

## 4.1 EXPERIMENTAL SETUP

Datasets, code, and scripts to reproduce all experimental results are available at `<link-provided-in-final-version>`.

**Knowledge graphs**. We used three commonly used benchmark datasets for evaluating KGE models: FB15K-237 (Toutanova & Chen, 2015), WNRR (Dettmers et al., 2018), and YAGO3-10 (Mahdisoltani et al., 2014). Each dataset is associated with a training, a validation and a test split. FB15K-237 and WNRR are designed to be harder benchmarks for link prediction. YAGO3-10 is not, but it is considerably larger. Dataset statistics are summarized in Table 5 in the appendix.

**KGE models**. We considered four popular, representative KGE models: TransE (Bordes et al., 2013) and DistMult (Yang et al., 2015) (basic translational and factorization models, resp.) as well as RotatE (Sun et al., 2019) and ComplEx (Trouillon et al., 2016) (SOTA translational and factorization models). RotatE and ComplEx are the methods of choice for low-cost embeddings with good prediction performance (Ruffinelli et al., 2020; Sun et al., 2019), and—with an increase in model size and/or training cost (Lacroix et al., 2018; Chen et al., 2021b)—can perform as well as SOTA models of other KGE model types such as the transformer-based HittER model (Chen et al., 2021a).

**KGE training**. We used LibKGE (Broscheit et al., 2020) for *STD training* (LP only) as a baseline and added MTT/MTR model training/evaluation. All KGE models were trained for a maximum of 200 epochs with early stopping on validation MRR checked every 10 epochs. We used cross-entropy as loss function, as it systematically outperformed other losses in most prior studies. We used *1vsAll* training with FB15K-237 and WNRR (to achieve good results) and *NegSamp* with YAGO3-10 (to scale to this larger dataset). Since we are interested in pre-trained KGE models, no information from downstream tasks is used for KGE model training and selection; e.g., the same KGE model is used for all downstream tasks in each experiment. In particular, models were selected w.r.t. performance

(MRR) on the validation data. Unless stated otherwise, models trained with STD training use the LP task, models trained with MTT training use MTR. Further improvements may be made by using downstream tasks during training (Aribandi et al., 2022); we leave such exploration to future work.

**KGE evaluation**. We evaluate KGE models with respect to each of the five graph-structure prediction tasks of Sec. 3 (LP, REL, DOM, NBE, NBR) using filtered MRR on test data. We also aggregate these metrics into the multi-task ranking MRR (MTR).

**KGE hyperparameters**. We closely follow the approach of the experimental study of Ruffinelli et al. (2020) to perform hyperparameter selection. We performed 30 random trials using SOBOL sampling (Bergstra & Bengio, 2012) over a large search space to tune several hyperparameters, e.g. regularization, embedding size, batch size, dropout, initialization, and task weights (each in $[0.1, 10.0]$, log scale). To keep our study feasible, we reduced the maximum batch and embedding size for larger datasets and expensive models. The full search space can be found in Table 8.

**Downstream tasks**. We collected or created data for 27 downstream tasks on FB15K-237 or YAGO3-10. This includes the datasets of Jain et al. (2021) for entity classification on FB15K-237 and YAGO3-10, which aims to predict the types of entities at different granularities. For regression, we use the datasets of Pezeshkpour et al. (2018) for YAGO3-10, which consist of temporal prediction tasks (e.g., the year an event took place), and the dataset of Huang et al. (2021) for node importance prediction. We also created several regression tasks for FB15K-237 from the multi-modal data of García-Durán et al. (2018) by predicting literals associated to entities (e.g., a date, a person's height, the rating of a movie). Datasets statistics are given in Tables 6 and 7 in the appendix.

**Downstream models**. We use scikit-learn (Pedregosa et al., 2011) using only the node embedding of the pre-trained KG model as input. For classification, we use multilayer perceptrons (MLP), logistic regression, KNN, and random forests. For regression, we use MLP and linear regression.

**Downstream training**. Each model was trained using 5-fold cross validation and selected based on mean validation performance across folds (see below). We then retrained the selected model on the union of the training and validation split (if present). To tune hyperparameters, we use 10 trials of random search with SOBOL sampling for each downstream model. The search space is given in Table 9. Note that we treat the choice of downstream model as a hyperparameter as well.

**Downstream evaluation**. For entity classification, we report *weighted F1*, as in Jain et al. (2021), aggregated across all classification tasks (denoted EC). For regression, we chose relative squared error (RSE) because it is interpretable and allows meaningful averaging across the different regression tasks (denoted REG). An RSE value of 1 is equivalent to the performance of a model that predicts the average of the dependent variable in the evaluation data; lower values are better. For each metric, we report the mean and standard deviation over 3 training runs of the downstream model.

**Downstream baselines**. We consider multiple baseline models to contextualize the results from pre-trained KGE models. In contrast to KGEs, the baselines are directly trained on the downstream task (i.e., no pre-training) and need to access the KG to perform predictions. We include KE-GCN (Yu et al., 2021), a recent GNN with state-of-the-art results for graph alignment and entity classification. For regression tasks, we use a linear layer after the final convolutional layer of KE-GCN.[2] We tune hyperparameters using 30 SOBOL trials (as for KGE models); the search space is shown in Table 9. For training, evaluation, and model selection, we follow the approach for our downstream models (e.g., 5-fold CV). We also consider selected SOTA results of other downstream models; see Sec. 4.5.

## 4.2 GRAPH-STRUCTURE PREDICTION

In Table 2, we report test MRR of all graph-structure prediction tasks from Table 1 for KGE models using standard training and link prediction for model selection (STD) and our proposed multi-task training and model selection (MTT).[3] Bold entries show best performance per metric and evaluation method. For easier comparison between STD and MTT, underlined entries highlight the best performance compared to the entry with the same corresponding KGE model on the same dataset, but that uses the other training method. The columns labeled *Downstream Tasks* are discussed in Sec. 4.3.

---

[2]In our experiments, this led to better performance than using a single dimensional output in the final convolution layer as done by Huang et al. (2021).

[3]Due to space constraints, we report results on WNRR in Table 11 in the appendix.

| | | | Graph-structure prediction (↑) | | | | | | Downstream tasks | |
| | | | LP | REL | DOM | NBE | NBR | MTR | EC (↑) | REG (↓) |
|---|---|---|---|---|---|---|---|---|---|---|
| *FB15K-237* | ComplEx | STD | **.347** | .805 | .098 | .011 | .041 | .200 | .844±.008 | .447±.051 |
| | | MTT | .331 | **.976** | .159 | .048 | **.672** | **.378** | .838±.002 | **.441±.050** |
| | DistMult | STD | .342 | .388 | .032 | .007 | .033 | .135 | .873±.009 | .551±.062 |
| | | MTT | .327 | .957 | **.160** | .034 | .670 | .371 | .862±.001 | .490±.052 |
| | RotatE | STD | .287 | .919 | .114 | .024 | .106 | .225 | .864±.005 | .618±.037 |
| | | MTT | .295 | .965 | .158 | .047 | .658 | .370 | **.886±.003** | .546±.081 |
| | TransE | STD | .300 | .900 | .111 | .018 | .049 | .213 | .881±.003 | .628±.045 |
| | | MTT | .291 | .963 | .155 | **.062** | .641 | .368 | .855±.003 | .991±.147 |
| | KE-GCN† | | – | – | – | – | – | – | .829±.526 | .501±.001 |
| *YAGO3-10* | ComplEx | STD | **.550** | .890 | .001 | .050 | .386 | .318 | .712±.008 | .589±.023 |
| | | MTT | .510 | .943 | .045 | .071 | .713 | .403 | .729±.003 | .539±.037 |
| | DistMult | STD | .539 | .877 | .004 | .069 | .434 | .330 | .738±.003 | .519±.019 |
| | | MTT | .538 | .941 | **.046** | .061 | **.740** | **.412** | .745±.007 | .476±.059 |
| | RotatE* | STD | .429 | – | – | – | – | – | .693±.002 | .745±.030 |
| | | MTT | .324 | .931 | .032 | .079 | .638 | .343 | .727±.004 | .593±.074 |
| | TransE* | STD | .490 | – | – | – | – | – | .741±.001 | .484±.053 |
| | | MTT | .263 | **.959** | .037 | **.080** | .617 | .330 | **.755±.003** | **.326±.022** |
| | KE-GCN† | | – | – | – | – | – | – | .700±.223 | .398±.008 |

\* Not evaluated on new graph-structure prediction tasks due to high cost.
† GCN-based model by Yu et al. (2021) trained directly on downstream tasks.

Table 2: Performance on test data of graph-structure prediction and downstream tasks with STD and MTT training, as well as KE-GCN. For graph-structure prediction, we report MRR (higher is better), for entity classification (EC) we report weighted F1 (higher is better), and for regression (REG) we show relative squared error (lower is better). Bold entries show best performance per task. Underlined entries show best performance between STD and MTT.

The results show that across all datasets and KGE models, STD training performed poorly on all graph-structure tasks, except LP and (often) REL. The performance for these tasks improved significantly with MTT training in almost all cases; these tasks have been introduced as auxiliary training objectives. This suggests that models trained solely using link prediction fail to capture graph structure more generally. Also note that MTT models had slightly lower performance on LP, but the decrease was often small and outweighted by significantly improved performance over the other tasks (often 2x–4x, up to 10x, depending on model, task and dataset). A notable exception was NBE, which is the only task that uses wildcard embeddings for relations. Here STD occasionally outperformed MTT (on YAGO-10 using DistMult and often on WNRR; see Tab. 11 in the appendix). Generally, however, MTT improved significantly on STD for graph structure prediction and can thus be used to improve KGE's ability to learn multiple graph tasks simultaneously.

## 4.3 Downstream Tasks

Table 2 also shows mean performance across all downstream tasks for each benchmark dataset. As before, bold entries show best performance per metric and evaluation method, and underlined entries facilitate performance comparisons across the different training approaches. We report performance for each individual downstream task in tables13 to 16 in the appendix.

The best overall downstream task performance across all KGE and KE-GCN models was achieved by MTT in all cases. The margin compared to STD was sometimes small (e.g., EC on FB15K-237) and sometimes large (e.g., REG on YAGO3-10). The margin compared to KE-GCN, which trains directly on each task, was large. Nevertheless, STD training occasionally performed better than MTT (e.g., on EC tasks for FB15K-237). This suggests that capturing a wider variety of graph structures does not necessarily translate to better downstream task performance. We explore this further in Section 4.4, where we consider subsets of the MTT tasks. Ultimately, we conclude that

| | | Graph-structure prediction (↑) | | | | | | Downstream tasks | |
|---|---|---|---|---|---|---|---|---|---|
| | | LP | REL | DOM | NBE | NBR | MTR | EC (↑) | REG (↓) |
| ComplEx | STD | **.347** | .805 | .098 | .011 | .041 | .200 | .844±.008 | .447±.051 |
| | MTT | .331 | **.976** | .159 | .048 | .672 | **.378** | .838±.002 | **.441±.050** |
| | w/o LP | .201 | .972 | .159 | **.065** | .676 | .356 | .822±.006 | .718±.040 |
| | w/o DOM | .302 | .967 | .153 | .023 | **.677** | .372 | **.883±.009** | .450±.031 |
| | w/o NBE | .308 | .967 | **.161** | .003 | **.677** | .371 | .874±.008 | .512±.038 |
| | w/o NBR | .299 | .971 | .155 | .034 | .487 | .331 | .870±.008 | .506±.059 |
| TransE | STD | .300 | .900 | .111 | .018 | .049 | .213 | .881±.003 | .628±.045 |
| | MTT | .291 | .963 | .155 | .062 | .641 | .368 | .855±.003 | .991±.147 |
| | w/o LP | .249 | .968 | .160 | .034 | .667 | .359 | .870±.000 | .456±.034 |
| | w/o DOM | .294 | .965 | .151 | .033 | .672 | .370 | .882±.002 | .515±.088 |
| | w/o NBE | .299 | .966 | .159 | .009 | .667 | .366 | .881±.004 | .466±.033 |
| | w/o NBR | .296 | .964 | .156 | .059 | .572 | .354 | .859±.002 | .603±.150 |

Table 3: Performance on FB15K-237 of graph-structure prediction and downstream tasks of STD and various forms of multi-task training of KGE models on test data. Metrics and format follow those of Table 2. The objective w/o LP is an MTT objective with all tasks in Table 1 except for LP.

the choice of pre-training objective clearly has an impact on downstream performance, although it is currently unclear how to make this choice.

Our results also suggest that—perhaps surprisingly—models with weaker performance during pre-training with both STD and MTT often performed competitively in downstream tasks and sometimes even outperformed models with stronger pre-training performance. For example, ComplEx considerably outperformed RotatE and TransE on FB15K-237 on LP and MTR, but both models outperformed ComplEx on the EC tasks for that dataset. Similar observations can be made about both EC and REG tasks on YAGO3-10. The REG tasks on FB15K-237 were an exception though; here higher performance during pre-training translated to better performance on downstream tasks. Generally, these results are problematic, as they suggest that LP and MTR are often inadequate to guide the choice of the KGE model class, a problem that needs further exploration in future work.

### 4.4 IMPACT OF TASK SELECTION AND MODEL SELECTION

Next, we explored the impact of task selection and, in particular, whether all proposed MTT tasks are beneficial. To keep computational costs feasible, we focused on FB15K-237 with ComplEx and TransE. We explored performance using STD, MTT, and MTT without either the LP, DOM, NBE, or NBR pre-training task. Our results are summarized in Tab. 3.

We found that for graph structure predictions, excluding a task generally led to lower performance on that task, as expected. It may also, however, lead to a boost in performance on other tasks. For example, the best NBE performance for ComplEx is obtained when LP is excluded.

For downstream tasks, we observe that the choice of training tasks can have a significant impact and that good choices differ between KGE models and downstream tasks. For example, compared to full MTT training, using a subset of tasks led to large improvements for ComplEx on EC and for TransE on REG. In both cases, as well as with TransE on EC, the best performance is obtained by removing one of the tasks during training. This reinforces our previous observation that including more tasks during pre-training does not necessarily lead to higher downstream performance, but it also provides more evidence that STD training is not enough for good downstream task performance. In fact, good models can be obtained without including the link prediction tasks: e.g., the best performance for TransE on REG was obtained when LP was excluded.

We also explored the impact of model selection methods. Table 4 reports performance on FB15K-237 of some KGE models using both training approaches across different types of model selection methods: selecting on LP (the standard approach), selecting on MTR and selecting directly on the metric used to evaluate the downstream task. We found that STD training performed best in combination with LP model selection. MTT performance on downstream tasks improved consistently

| | | | Selection Method | | | | | |
|---|---|---|---|---|---|---|---|---|
| | | | EC - Weighted F1 (↑) | | | REG - RSE (↓) | | |
| | | | LP | MTR | Weighted F1 | LP | MTR | RSE |
| *FB15K-237* | ComplEx | STD | .844 | .830 | .850 | .447 | .654 | .437 |
| | | MTT | .858 | .838 | .827 | .394 | .441 | .393 |
| | DistMult | STD | .873 | .825 | .846 | .550 | .677 | .539 |
| | | MTT | .865 | .861 | .864 | .471 | .489 | .476 |

Table 4: Performance on FB15K-237 downstream tasks for different KGE model training (STD/MTT) and selection approaches (LP/MTR/weighted F1/RSE). Weighted F1/RSE use downstream tasks data for model selection.

when using LP instead of MTR for model selection, however. Model selection with the downstream task metric provides only marginal benefits for both STD and MTT and can in fact be detrimental, likely due to overfitting on validation data. This indicates that model selection without information about downstream tasks—i.e., using LP or MTR—is suitable. The combination that performed best in our study was MTT training and LP model selection.

Overall, we found that full MTT training and MTR for model selection (as used in our main results of Tab. 2) was a suitable choice, but further improvements are possible by dataset-, model- and task-specific choices of pre-training task and validation objective.

### 4.5 COMPARISON TO TASK-SPECIFIC MODELS

We compared the performance of a pre-trained ComplEx model (using MTT) to best results for additional downstream tasks from the literature. These prior results were obtained by task-specific models and were not reproduced by us; see Sec. A.3 for a description of tasks and detailed results. We found that in most cases, this pre-trained ComplEx model did not reach the performance of SOTA task-specific models (which in some cases leveraged additional information). More exploration is needed to whether and when pre-trained KGE models are preferable (e.g., as in the tasks of Tab. 2) and on the effectiveness-cost trade-off of alternative approaches.

## 5 CONCLUSION

In this work, we explored methods to pretrain KGE models for tasks beyond link prediction. First, we showed empirically that commonly trained KGE models fail at basic graph-structure prediction tasks and proposed a novel multi-task training and ranking approaches. These multi-task KGE models led to substantially better performance, i.e, their embeddings captured more information about graph structure. Second, we explored downstream task performance for a number of entity classification and regression tasks. Here multi-task training generally led to the best overall performance, but the margin was sometimes small. Our ablation studies suggest that pre-training can be further improved by a data- and model-specific selection of both pre-training tasks and model selection metric. Generally, more research is needed on how to make these choices and, more generally, on the relation between pre-training KGE models and their general suitability for downstream applications.

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

| Dataset | Entities | Relations | Train | Validation | Test | EC Tasks | REG Tasks |
|---|---|---|---|---|---|---|---|
| FB15K-237 | 14 505 | 237 | 272 115 | 17 535 | 20 466 | 4 | 10 |
| YAGO3-10 | 123 182 | 37 | 1 079 040 | 5 000 | 5 000 | 8 | 5 |
| WNRR | 40 559 | 11 | 86 835 | 3 034 | 3 134 | 0 | 0 |

Table 5: Statistics of benchmark datasets for pre-training KGEs, including number of entity classification (EC) tasks and regression (REG) tasks.

## A APPENDIX

### A.1 DATASET STATISTICS

| Benchmark | Name | Num. Classes | Train | Validation | Test |
|---|---|---|---|---|---|
| FB15K-237 | Entity Type | 3 (person, org, . . .) | 6 719 | – | 1 680 |
| | Profession | 5 (artist, writer, . . .) | 2 537 | – | 635 |
| | Organization Type | 4 (NGO, musical_org, . . .) | 342 | – | 86 |
| | Writer Type | 2 (journalist, poet, . . .) | 136 | – | 34 |
| YAGO3-10 | Entity Type | 4 (person, org, . . .) | 69 592 | – | 17 398 |
| | Player Type | 5 (soccer, hockey, . . .) | 33 928 | – | 8 483 |
| | Profession | 5 (artists, politician, . . .) | 14 480 | – | 3 621 |
| | Writer Type | 7 (poet, novelist, . . .) | 4 870 | – | 1 218 |
| | Scientist Type | 10 (physicist, biologist, . . .) | 2 041 | – | 511 |
| | Organization Type | 4 (institution, party, . . .) | 1 248 | – | 312 |
| | Artists Type | 5 (painter, sculptor, . . .) | 520 | – | 130 |
| | Waterbody Type | 5 (lake, ocean, . . .) | 195 | – | 49 |

Table 6: Statistics of datasets for entity classification downstream tasks used to evaluate pre-trained KGEs. All datasets were created by Jain et al. (2021), they are split into trainining and test only and each consists of predicting entity types at different levels of the entity hierarchy.

| Benchmark | Name | Task | Train | Validation | Test |
|---|---|---|---|---|---|
| FB15K-237 | Node Importance | (Entity, Wikipedia page views) | 9 877 | 1 380 | 2 823 |
| | Birth Year | (People, year of birth) | 3 538 | 442 | 444 |
| | Latitude | (Location, latitude) | 2 568 | 321 | 322 |
| | Longitude | (Location, longitude) | 2 560 | 320 | 322 |
| | Person Height | (Person, height in meters) | 2 295 | 287 | 288 |
| | Size Area | (Location, area) | 1 731 | 216 | 218 |
| | Population | (Location, population) | 1 543 | 193 | 193 |
| | Film Release Year | (Film, release year) | 1 493 | 186 | 188 |
| | Org Year Founded | (Organization, year founded) | 985 | 123 | 124 |
| | Film Rating | (User, rated film 1 to 100) | 591 | 73 | 75 |
| YAGO3-10 | Born on Year | (Person, year of birth) | 60 409 | – | 6 730 |
| | Created on Year | (Entity, year of creation) | 23 896 | – | 2 638 |
| | Died on Year | (Person, year of death) | 13 582 | – | 1 513 |
| | Destroyed on Year | (Entity, year of end/death) | 1 630 | – | 186 |
| | Happened on Year | (Event, year it took place) | 749 | – | 73 |

Table 7: Statistics of datasets for regression downstream tasks used to evaluate pre-trained KGEs. YAGO3-10 datasets were created by Pezeshkpour et al. (2018). All FB15K-237 datasets were created by us, except the node importance task, which was created by Huang et al. (2021).

## A.2 EXPERIMENTAL SETTINGS

| Hyperparameter | Values |
| --- | --- |
| Embedding size[†] | $\{128, 256, 512\}$ |
| Training type | $\{$NegSamp (YAGO3-10), 1vsAll (FB15K, WNRR)$\}$ |
| Task Weights (MTT) | $[0.1, 10]$, log scale |
|    No. subject samples (NegSamp) | $[1, 10000]$, log scale |
|    No. object samples (NegSamp) | $[1, 10000]$, log scale |
| Optimizer | $\{$Adam, Adagrad$\}$ |
|    Batch size[*] | $\{128, 256, 512, 1024$(except on YAGO3-10)$\}$ |
|    Learning rate | $[10^{-4}, 1]$, log scale |
|    LR scheduler patience | $[0, 10]$ |
| $L_p$ regularization | $\{$L1, L2, L3, None$\}$ |
|    Entity emb. weight | $[10^{-20}, 10^{-5}]$ |
|    Relation emb. weight | $[10^{-20}, 10^{-5}]$ |
|    Frequency weighting | $\{$True, False$\}$ |
| Embedding normalization (TransE) | |
|    Entity | $\{$True, False$\}$ |
|    Relation | $\{$True, False$\}$ |
| Dropout | |
|    Entity embedding | $[0.0, 0.5]$ |
|    Relation embedding | $[0.0, 0.5]$ |
| Embedding initialization | $\{$Normal, Unif, XvNorm, XvUnif$\}$ |
|    Std. deviation (Normal) | $[10^{-5}, 1.0]$ |
|    Interval (Unif) | $[-1.0, 1.0]$ |
|    Gain (XvNorm) | $1.0$ |
|    Gain (XvUnif) | $1.0$ |

[†] For RotatE, embedding size is fixed 128 on WNRR and set to either 128 or 256 for YAGO3-10. For Transe, this is set to either 128 or 256 for FB15K-237 and fixed to 128 for WNRR and 1024 for YAGO3-10.

[*] For RotatE, batch size is fixed to 256 in YAGO3-10 and to 128 on FB15K-237 and WNRR. For Transe, this is set to either 128 or 256 on YAGO3-10.

Table 8: Hyperparameter search space for pre-training KGE models. Restrictions specific to RotatE and TransE are due to high memory and runtime cost when training these models.

| Model | Hyperparameter | Values |
| --- | --- | --- |
| MLP | Hidden Layer Sizes | $\{(100, ), (10, ), (100, 100), (10, 10)\}$ |
| | Alpha | $[0.00001, 0.001]$ |
| | Learning Rate Init | $[0.001, 0.01]$ |
| | Solver | $[Adam, LBFGS]$ |
| Logistic Regression | C | $[100, 100000]$ |
| KNN | n_neighbors | $[1, 10]$ |
| Random Forest | num_estimators | $[10, 50, 100, 200]$ |
| Linear Regression | Alpha | $[0.00001, 0.001]$ |
| KE-GCN | Dimension | $\{16, 32, 64\}$ |
| | Additional Layers | $\{0, 1, 2\}$ |
| | Learning Rate | $\{0.001, 0.005, 0.01, 0.05, 0.1\}$ |
| | Alpha | $\{0.3, 0.5\}$ |

Table 9: Hyperparameter search space for training downstream models. All hyperparameters except those of KE-GCN follow the semantics by scikit-learn.

## A.3 COMPARISON TO SOTA RESULTS

In Table 10, we compare downstream task performance of a pre-trained ComplEx model using the MTT approach with state-of-the-art task-specific models from the literature. Note that we did not train these models ourselves and experimental setups used in prior work may be very different from ours. In these experiment, we did not reach state-of-the-art performance with KGE models. Aside from training directly for the task, this may be because (i) some of the datasets (AIFB, MUTAG) are very small so that KGE models do not learn much before overfitting, (ii) some knowledge graphs are multi-modal (MDGENRE, DMGFULL), but we do not leverage this information[4] On MDGENRE, however, the additional modalities do not play a significant role, as not only does our pre-trained KGE achieve comparative performance, but so do the non-multimodal baselines from Bloem et al. (2021). This is not the case with the DMGFULL datasets, however, which contains a significantly higher number of triples from different modalities.

| Dataset | Task | Metric | Their model | Their performance | Ours (Selection method) |
|---|---|---|---|---|---|
| MDGENRE* | EC | Accuracy | Features | 0.66 | 0.66±0.01 (MTR) 
 0.68±0.00 (Acc) |
| DMGFULL* | EC | Accuracy | MR-GCN | **0.76** | 0.51±0.00 (MTR) 
 0.67±0.02 (Acc) |
| FB15K† | NIE | NDCG@100 | RGTN | **0.95** | 0.41±0.00 (MTR) 
 0.42±0.00 (NDCG) |
| | | Spearman | RGTN | **0.82** | 0.75±0.00 (MTR) 
 0.76±0.00 (Spearman) |
| MUTAG‡ | EC | Accuracy | CompGCN | **0.85** | 0.75±0.02 (MTR) 
 0.75±0.01 (Acc) |
| AIFB§ | EC | Accuracy | R-GCN | **0.95** | 0.88±0.00 (MTR) 
 0.88±0.00 (Acc) |

\* Bloem et al. (2021), †Huang et al. (2021), ‡Vashishth et al. (2020), §Schlichtkrull et al. (2018)

Table 10: Comparison of entity classification (EC) and node importance estimation (NIE) between best previously published models (directly trained on task) and pre-trained KGEs (MTT, selection with MTR or downstream metric).

---

[4]There are multi-modal KGE models, however, e.g. (Pezeshkpour et al., 2018).

A.4 ADDITIONAL EXPERIMENTAL RESULTS

| | | | *Graph-structure prediction (↑)* | | | | | | *Downstream tasks* | |
| | | | LP | REL | DOM | NBE | NBR | MTR | EC (↑) | REG (↓) |
|---|---|---|---|---|---|---|---|---|---|---|
| *WNRR* | ComplEx | STD | **.474** | .782 | .001 | .150 | .578 | .354 | – | – |
| | | MTT | .460 | .833 | .024 | .181 | **.791** | .416 | – | – |
| | DistMult | STD | .447 | .767 | .001 | .169 | .605 | .356 | – | – |
| | | MTT | .435 | **.897** | **.025** | .190 | .781 | **.417** | – | – |
| | RotatE | STD | .460 | .794 | .001 | **.210** | .713 | .398 | – | – |
| | | MTT | .422 | .874 | .024 | .152 | .790 | .408 | – | – |
| | TransE | STD | .174 | .707 | .001 | .119 | .307 | .212 | – | – |
| | | MTT | .175 | .837 | .021 | .109 | .747 | .327 | – | – |

Table 11: Performance on graph-structure prediction and downstream tasks of STD and multi-task KGE models as well as KE-GCN on test data. For graph-structure prediction, we report MRR (higher is better), for entity classification (EC) we report weighted F1 (higher is better), and for regression (REG) we show relative squared error (lower is better). Bold entries show best performance per task. Underlined entries show best performance between STD and MTT training.

| | | *Average training epoch time in seconds* | | |
| | | FB15K-237 | YAGO3-10 | WNRR |
|---|---|---|---|---|
| ComplEx | STD | 04.92 | 097.88 | 2.32 |
| | MTT | 10.83 | 137.13 | 8.13 |
| DistMult | STD | 04.29 | 095.82 | 02.10 |
| | MTT | 09.27 | 222.37 | 10.98 |

Table 12: Average training epoch time in seconds over first 5 epochs of best models with STD and MTT training. All tests were done with an 11th gen. Intel Core i7-11700K, 64GB of RAM and an NVIDIA GeForce RTX 3090.

| | | FB15K-237 | | |
| | | Entity Classification (Weighted F1 - higher is better) | | |
| | | Type | Profession | Organization | Writer |
|---|---|---|---|---|---|
| ComplEx | STD | .986±.001 | .808±.011 | .921±.021 | .661±.000 |
| | MTT | .986±.000 | **.811±.007** | .900±.000 | .656±.000 |
| DistMult | STD | .984±.000 | **.811±.007** | .912±.009 | .785±.020 |
| | MTT | .986±.000 | .803±.005 | .929±.000 | .729±.000 |
| RotatE | STD | .983±.000 | .798±.007 | .895±.012 | .781±.000 |
| | MTT | .987±.000 | .797±.010 | .933±.004 | .828±.000 |
| TransE | STD | .987±.001 | .794±.007 | .935±.005 | **.810±.000** |
| | MTT | **.988±.001** | **.811±.003** | **.937±.005** | .683±.005 |
| KE-GCN | | **.988±.000** | .738±.000 | .906±.002 | .685±.020 |

Table 13: Weighted F1 on test data of downstream classifiers (MLP, Logistic Regression, KNN and Random Forest) that use pre-trained KGE embeddings as input to solve entity classification tasks about entities in FB15K-237; and KE-GCN (Yu et al., 2021), a GCN that trains directly on the downstream data. Bold entries show best performance per task. Underlined entries show best performance per task compared to same KGE model and same downstream model. Datasets are sorted by decreasing size of the training set from left to right.

| | | YAGO3-10 | | | | | | |
| | | Entity Classification (Weighted F1 - higher is better) | | | | | | |
| | Type | Player | Profession | Writer | Scientist | Organization | Artist | Waterbody |
|---|---|---|---|---|---|---|---|---|
| ComplEx STD | .994±.000 | .918±.001 | .753±.004 | .575±.006 | .518±.013 | .789±.005 | .480±.018 | .673±.015 |
| MTT | **.996±.000** | .918±.002 | .761±.003 | .607±.000 | .538±.002 | .867±.015 | .476±.005 | .664±.000 |
| DistMult STD | .994±.000 | .919±.001 | .764±.003 | .577±.000 | .529±.003 | .814±.011 | .535±.007 | **.738±.000** |
| MTT | **.996±.000** | .919±.001 | **.790±.001** | .618±.023 | .536±.015 | **.883±.005** | .496±.014 | .722±.000 |
| RotatE STD | .974±.000 | .918±.002 | .709±.002 | .606±.000 | .545±.000 | .730±.013 | .470±.000 | .593±.000 |
| MTT | .993±.000 | .914±.001 | .761±.000 | .628±.000 | .548±.013 | .816±.007 | .531±.000 | .625±.013 |
| TransE STD | .992±.000 | .920±.001 | .762±.000 | .623±.000 | **.630±.000** | .833±.000 | .499±.013 | .670±.000 |
| MTT | .995±.000 | **.923±.001** | .788±.003 | **.666±.000** | .600±.014 | .853±.008 | **.543±.000** | .673±.000 |
| KE-GCN | **0.996±0.000** | .896±.001 | .709±.000 | .582±.005 | .610±.006 | .853±.006 | .463±.014 | .488±.014 |

Table 14: Weighted F1 on test data of downstream classifiers (MLP, Logistic Regression, KNN and Random Forest) that use pre-trained KGE embeddings as input to solve entity classification tasks about entities in YAGO3-10; and KE-GCN (Yu et al., 2021), a GCN that trains directly on the downstream data. Bold entries show best performance per task. Underlined entries show best performance per task compared to same KGE model and same downstream model. Datasets are sorted by decreasing size of the training set from left to right.

| | | Node Imp. | Birth Year | Latitude | Longitude | Person Height | Size Area | Population | Film Rel. Year | Date Founded | Film Rating |
|---|---|---|---|---|---|---|---|---|---|---|---|
| ComplEx | STD | .870±.048 | .601±.239 | .172±.013 | .089±.010 | .678±.010 | .234±.018 | .442±.071 | .156±.016 | .494±.042 | .736±.046 |
| | MTT | **.771±.014** | **.302±.048** | .211±.010 | .091±.002 | **.656±.015** | .457±.266 | .652±.033 | **.106±.004** | .440±.083 | .727±.029 |
| DistMult | STD | .807±.023 | .844±.042 | .182±.031 | .088±.005 | .669±.003 | .412±.318 | .914±.093 | .152±.003 | .627±.036 | .813±.062 |
| | MTT | .944±.066 | .722±.091 | .255±.016 | .094±.008 | .674±.006 | **.089±.057** | .498±.075 | .133±.004 | .584±.044 | .907±.148 |
| RotatE | STD | .877±.026 | .906±.027 | .520±.044 | .289±.032 | .657±.000 | .911±.000 | **.364±.125** | .175±.007 | .681±.048 | .798±.059 |
| | MTT | .846±.005 | .822±.067 | .311±.028 | .166±.011 | .722±.020 | .428±.219 | .764±.248 | 0.150±.011 | .472±.134 | .776±.062 |
| TransE | STD | .919±.000 | .756±.110 | .158±.003 | **.079±.005** | .730±.021 | .432±.037 | 1.702±.055 | .171±.027 | .551±.123 | .778±.064 |
| | MTT | .833±.015 | .668±.057 | **.102±.015** | .041±.002 | .780±.010 | 5.902±.946 | .830±.347 | .163±.007 | **.343±.060** | .740±.009 |
| KE-GCN | | .804±.005 | .376±.035 | .218±.023 | .113±.003 | .748±.002 | .754±.0180 | .664±.051 | .144±.008 | .498±.034 | **.691±.009** |

*FB15K-237*
*Regression (RSE - lower is better)*

Table 15: Relative squared error (RSE) on test data of downstream models (MLP and Linear Regression) that use pre-trained KGE embeddings as input to solve regression tasks about entities in FB15K-237; and KE-GCN (Yu et al., 2021), a GCN that trains directly on the downstream data. Bold entries show best performance per task. Underlined entries show best performance per task compared to same KGE model and same downstream model. Models with RSE above 1 are considered unsatisfactory. Datasets are sorted by decreasing size of the training set from left to right.

_YAGO3-10_
_Regression (RSE - lower is better)_

| | | Born on Date | Created on Date | Died on Date | Destroyed on Date | Happened on Date |
|---|---|---|---|---|---|---|
| ComplEx | STD | .519±.001 | .672±.033 | .555±.014 | .872±0.060 | .324±.006 |
| | MTT | .440±.011 | .619±.015 | .520±.031 | .729±.036 | .376±.093 |
| DistMult | STD | .432±.013 | .612±.024 | .466±.025 | .773±.004 | .311±.030 |
| | MTT | .333±.013 | **.560**±.012 | .398±.018 | .725±.018 | .365±.068 |
| RotatE | STD | .691±.019 | .828±.025 | .849±.000 | 1.040±.091 | .314±.014 |
| | MTT | .412±.011 | 1.097±.266 | .480±.022 | .750±.060 | .224±.008 |
| TransE | STD | .365±.017 | .687±.034 | .376±.028 | .617±.063 | .376±.125 |
| | MTT | .268±.013 | .619±.034 | .294±**.013** | **.357**±.041 | **.091**±**.008** |
| KE-GCN | | **.256**±**.009** | .611±.008 | .299±.011 | .657±.045 | .167±.001 |

Table 16: Relative squared error (RSE) on test data of downstream models (MLP and Linear Regression) that use pre-trained KGE embeddings as input to solve regression tasks about entities in YAGO3-10; and KE-GCN (Yu et al., 2021), a GCN that trains directly on the downstream data. Bold entries show best performance per task. Underlined entries show best performance per task compared to same KGE model and same downstream model. Models with RSE above 1 are considered unsatisfactory. Datasets are sorted by decreasing size of the training set from left to right.

