# OpenReview forum: "Beyond Link Prediction: On Pre-Training Knowledge Graph Embeddings"
_ICLR.cc/2023/Conference — Submitted to ICLR 2023_

### Official Review · Reviewer_x7Bp · 2022-10-24

**Confidence:** 4
**Correctness:** 3
**Technical Novelty And Significance:** 2
**Empirical Novelty And Significance:** 2
**Recommendation:** 5

**Clarity, Quality, Novelty And Reproducibility:**

The paper is clearly written and the representation is good. However, the model behavior should be further discussed or explained.
This paper contributes little technical novelty. Albeit some observations are interesting, the proposed method does not show enough effectiveness.



**Details Of Ethics Concerns:**

No ethics concerns as far as I can see.

**Strength And Weaknesses:**

Strength:
1.	Some observations are interesting, e.g., KGE performs poorly on graph-structure prediction tasks other than link prediction.
2.	The paper is clearly written.

Weakness:
1.	The results are not stunning. The multi-task training does not lead to consistently better performance on all graph-structure prediction tasks. The performance improvement on downstream tasks is not effective.
2.	Some experimental settings are not rigorously designed. E.g., MTT should test on unseen tasks. The experimental results could be further discussed or explained, i.e., the variation of model behavior under different datasets/settings.


**Summary Of The Paper:**

The paper proposed a multi-task training strategy for improving the performance of the knowledge graph embedding (KGE) model in downstream tasks, e.g., entity classification or regression. In addition to the conventional link prediction training of KGE, the proposed model is pre-trained with 4 extra tasks, e.g., relation prediction, entity neighboring, etc. The experiments on FB15K-237, YAGO3-10, and WRNN show the multi-task trained model achieves better MRR on the pretraining tasks but lags behind LP than the single training (as expected). The performance on downstream tasks are marginally better for certain KGE model, e.g., RotatE on FB15K-237. Overall, the performance of graph-structure prediction tasks (e.g., NBE) is not stunning as dedicatedly trained. The performance of downstream tasks is also limited.

**Summary Of The Review:**

1.	The results of applying multi-task training are not stunning at all. The model trained on the single link prediction task (STD) still has the best performance on the link prediction task (Table 3). And the model trained on extra tasks (MTT) no doubt performs well on the extra tasks. I would suggest authors train models ablative, i.e., for the unseen test tasks. In addition, the performance improvements on downstream tasks are also limited. E.g., On the FB15K-237 dataset, the STD strategy excels the proposed MTT for nearly all settings of EC (except for RotatE). In this regard, the effectiveness of the proposed multi-task training strategy is nor promising.

2.	The domain and range predictions seem not widely used tasks for KGE. Are there any reasons for choosing these two tasks? E.g., do they have any potential for benefiting downstream applications?

3.	As the SOTA performances of KGC are achieved by pretrained LM-based models, such as KG-BERT (Yao et al., 2019), KGT5 (Saxena et al., ACL’22), and KG-S2S (Chen, et al., COLING’22). The applicability of the proposed MTT strategy should be further discussed.

4.	It is not clear how the wildcard entities (MTT part) approach the max-aggregation (MTR part).  The max-aggregation search for the best relation over entire entity set, while the wildcard entities make use of an abstract "wildcard" embedding to get the result. Although using the wildcard entity would be easier as it reduces the computational cost for searching, but I believe the two methods The methodologies for the two methods are conceptionally different and may have different results.

5.	Why STD has better results than the dedicated MTT on NBE task (0.21 v.s. 0.152, Table 3) needs further explanation.

Minor comments:
(1) Authors should give the description of abbreviation STD (i.e., the model trained with single task, i.e., LP.) when it is first introduced in Section 4.1.
(2) .. replace wildcards by their the corresponding .. -> remove the (page 5)

---

> ### Author Response · Authors · 2022-11-18
> **Reply to your concerns**
>
> > Concern: multi-task training does not lead to consistently better performance on all graph-structure prediction tasks.
>
> Yes. The performance generally slightly dropped on LP and it generally improved on REL, DOM, and NBR. For NBE, the situation was mixed (esp. on WNRR). We now state this more clearly in the paper. In all cases, however, the MTR results are much better for multi-task training, i.e., it is generally very effective for graph-structure prediction.
>
> > Concern: the performance improvement on downstream tasks is not effective.
>
> > Concern: On the FB15K-237 dataset, the STD strategy excels the proposed MTT for nearly all settings of EC (except for RotatE).
>
> > Concern: the effectiveness of the proposed multi-task training strategy is not promising.
>
> > Concern: Why STD has better results than the dedicated MTT on NBE task (0.21 v.s. 0.152, Table 3) needs further explanation.
>
> We partly agree. First, the improvement of the best results for EC were .881->.886 (FB) and .741->.755 (Yago). For REG, .447->.441 (FB) and .484->.326 (Yago). Especially on FB, the improvements are small. Second, these are aggregated results; the performance improvement can be larger for specific tasks (see appendix). Finally, the ablation study on model selection as well the newly-added study on task selection suggests that additional improvements are possible. More research is needed on how to make these choices, however. We expanded Sections 4.2, 4.3 and 4.4 to provide further discussion and insights.
>
> > Concern: MTT should test on unseen tasks.
>
> We have added Table 3 with ablation runs and discuss this in section 4.4.
>
> > Concern: The experimental results could be further discussed or explained.
>
> We've completely revised the discussion in the experimental section and added an experiment on task selection to provide further analysis and insight. We also greatly appreciate suggestions on how to further improve the discussion.
>
> > Question: Are there any reasons for choosing domain and range prediction tasks?
>
> These tasks test whether the model understands the "active domain" of a relation (e.g., that livesIn is a relation between persons and locations).
>
> We expanded the discussion on p. 4 accordingly. Quoting: " We chose this particular set of tasks because they are simple, they capture basic information about the graph structure beyond link prediction, and they only have one prediction target (an entity or a relation). The latter property allows efficient pre-training and evaluation, as discussed below. For this reason, we exclude tasks such as entity-pair prediction (denoted $(?,p,?)$ in our notation) or reconstruction (denoted $(?,?,?)$). In our experimental study, we also found that the exclusion of some of the above pre-training tasks (e.g., LP) can further improve downstream task performance. The optimal choice of tasks depends on dataset, KGE model, and downstream task, however. We leave the exploration of task selection as well as on exploring additional pre-training tasks to future work."
>
> > Concern: lack of discussion on SOTA LM-based link prediction models (e.g., KG-BERT, KGT5, KG-S2S).
>
> Our focus is on exploring KGE models for tasks beyond link prediction. We agree LM-based approaches might also be useful in such settings, but this is beyond the scope of this work.
>
> As a side note, it's not clear whether LM-based models are generally preferable for link prediction. E.g., our FB15k-237 models have better link prediction performance than KG-S2S. Likewise, KGT5 did not outperform ComplEx in the study of Saxena et al. (ACL’22), but an ensemble of KGT5 and ComplEx was very beneficial.
>
> > Concern: STD use of max-aggregation and MTT's use of wildcard embeddings are different approaches that may have different results.
>
> STD does not provide wildcard embeddings so that some form of aggregation needs to be performed (how else could we address these tasks with such models?). Max-aggregation is a natural choice as larger scores are meant to indicate higher model confidence.
>
> When training MTT models, we introduce wildcard embeddings instead because otherwise (1) training costs would be excessive and (2) prediction costs would be large (as for STD with max-aggregation, which is much more costly). Note that we cannot use wildcard embeddings with STD as they need to be trained.

---

### Official Review · Reviewer_1Rac · 2022-10-25

**Confidence:** 3
**Correctness:** 3
**Technical Novelty And Significance:** 3
**Empirical Novelty And Significance:** 3
**Recommendation:** 6

**Clarity, Quality, Novelty And Reproducibility:**

This paper is well written and the motivation of this paper is clear. While there are some minor weaknesses in the work, I feel that the research problem is well-founded and the authors have done a good job overall.
Authors use a well-known KGE framework for implementing the experiments in this paper but do not provide their code, so reproducibility takes a hit.


**Strength And Weaknesses:**

Strengths

+ Motivation for the work is well-founded in literature and concretely defined in the work.
+ Authors introduce several new basic graph structure prediction tasks (Neighborhood and Domain prediction). They also evaluate existing models on these tasks showing the poor generalizability of current KGE methods to such tasks.
+ Further, this work proposes an MTT objective that includes the well-known Link and Relation Prediction tasks in conjunction with other graph structure prediction tasks introduced in this work.
+ Evaluation shows that MTT training does help improve the existing KGE model's performance on graph structure prediction tasks and downstream tasks.

Weaknesses / Questions

+ Authors should provide an ablation study in order to better understand the role of each task introduced in MTT.
+ In my opinion, this method loses the Efficiency vs Performance tradeoff. It takes 2x-4x (worst) more time to train but seems to marginal gain in performance on downstream tasks.
+ It would have been interesting to see the results of KE-GCN on basic graph structure prediction tasks as well (treating these tasks as downstream and training on them).
+ Instead of YAGO3-10, why didn't the authors use one of the [OGB](https://ogb.stanford.edu/docs/linkprop/) datasets for large-scale experiments? I would like to know if this is a conscious design choice by the authors.
+ "We also use the so-obtained wildcard embeddings for prediction in the same fashion; e.g., we set $$s(s, r^{'}, *) = s(s, r^{'}, any_{O})$$". I am not sure which prediction the authors are talking about here.
+ Please provide the code used for all the experiments.

**Summary Of The Paper:**

This paper aims to explore the suitability of various Knowledge Graph Embeddings (KGE) for graph structure prediction tasks such as entity and relation neighborhood prediction, type/range of a given relation, and relation prediction. This text also investigates the performance of these models on several downstream tasks. Building upon these two investigations, the authors then propose a Multi-Task Training (MTT) objective to potentially improve the performance on graph structure prediction tasks and downstream applications without further task-specific training.

**Summary Of The Review:**

Overall, I feel this is a good piece of work with a well-known and well-founded research problem. Authors have done extensive evaluation (with some minor weaknesses) to explore the use of Multi-Task Training objectives to improve the generalizability of Knowledge Graph Embeddings methods. To that extent, I vote to accept the paper.

---

> ### Author Response · Authors · 2022-11-18
> **Reply to your concerns**
>
> > Suggestion: Authors should provide an ablation study on task selection.
>
> We added an ablation study (Sec. 4.4). Note that the computational cost of such experiments is very high; we thus focused on FB15k-237 with ComplEx and TransE.
>
> > Concern: MTT loses the efficiency vs performance trade-off.
>
> We partly agree. First, the improvement of the best results for EC were .881->.886 (FB) and .741->.755 (Yago). For REG, .447->.441 (FB) and .484->.326 (Yago). Especially on FB, the improvements are small. Second, these are aggregated results; the performance improvement can be larger for specific tasks (see appendix). Third, training cost indeed increases with MTR. But using MTT models for graph-structure prediction tasks is much cheaper (due to the use of any embeddings) and using them in downstream tasks is not more costly. Finally, the ablation study on model selection as well the new study on task selection suggests that additional improvements are possible. More research is needed on how to make these choices, however.
>
> > Suggestion: KE-GCN's performance on graph structure tasks.
>
> Training KE-GCN (or generally relational GNNs such as R-GCN) for link prediction is extremely costly and there is currently no experimental evidence that such models outperform KGE models. More recent work focuses on more suitable GNN architectures for link prediction. In particular, NBFnet [1] provides very good link prediction results, but it is not immediately clear how to use it for graph-structure prediction and it does not support or provide embeddings for other downstream tasks.
>
> > Question: was it a concious choice to not use OGB datasets?
>
> Yes. Most OGB datasets are either undirected graphs or have only a single relation type, whereas we focus on knowledge graphs. Out of the three KGs provided by OGB (mag, wikikg2, biokg), only mag is is associated with a downstream task. It is only one downstream task and the dataset is large, however. We considered FB15K-237 and Yago3-10 more suitable for our study.
>
> > Question: What is meant with "We also use the so-obtained wildcard embeddings for prediction in the same fashion; e.g., we set"?
>
> We clarified this in the paper. Generally, when an MTT model is used for a graph-structure prediction task such as (s,?,*), we use score(s,r,any_O) instead of max_o score(s,r,o) to obtain the score of candidate answer r.
>
> > Concern: code is not available and reproducibility takes a hit.
>
> We did not include code for anonymity reasons, but it will be made publicly available (including datasets and scripts to reproduce all experiments) once the paper is published. For now, we'll provide an anonymized (but not yet fully documented) snapshot of our code in a separate comment visible only to reviewers.

---

### Official Review · Reviewer_RUJy · 2022-10-25

**Confidence:** 4
**Correctness:** 3
**Technical Novelty And Significance:** 2
**Empirical Novelty And Significance:** 2
**Recommendation:** 5

**Clarity, Quality, Novelty And Reproducibility:**

Overall, the work is of acceptable quality and originality. But, there is still room for improvement in writing:

* "Knowledge graph embeddings (KGE) models provide low-dimensional representations of the entities and relations in a knowledge graph (KG)." -> Knowledge graph embedding (KGE) models provide low-dimensional representations of entities and relations.

* "..., STD training approach performs poorly on all graph-structure tasks" -> ..., the STD training approach performs poorly on all graph-structure tasks

* "The performance for these tasks is significantly increased using MTT training." -> The performance of these tasks is significantly improved by using MTT training.


**Strength And Weaknesses:**

This paper presents some "interesting" findings regarding KG embedding models and applications. It would motivate us to reconsider the suitability and generalizability of KG embedding models. However, I have the following concerns.

First, as a KG embedding expert, the finding does not surprise me that a model trained on the link prediction task performs poorly on other structure prediction tasks. There is a significant gap between the training and evaluation tasks in this experiment. It is also not surprising that multi-task learning can improve the model's performance on some structure prediction tasks, because, just as stated by the authors in Section 4.2, "these tasks have been introduced as auxiliary training objectives". So, I think this paper is not so attractive to the KG embedding researchers, although I appreciate the authors' hard work. It lacks in-depth analysis of the reasons behind the performance.

Second, the selected four KG embedding models in the empirical study, i.e., TransE, DistMult, ComplEx, and RotatE, are old. I think these four models could not adequately represent the large and fast-growing family of KG embedding models. In Section 4.1, the authors state that ComplEx is the SOTA factorization model. But in my view, the SOTA factorization-based KG embedding model is TuckER [1]. Also, some recent GNN-based [2] or Transformer-based KG embedding models are ignored in this study. So, a natural question is, whether the recent KG embedding models still suffer from the claimed issue in this paper.

[1] Ivana Balazevic, Carl Allen, and Timothy Hospedales. 2019. TuckER: Tensor Factorization for Knowledge Graph Completion. In Proceedings of the 2019 Conference on Empirical Methods in Natural Language Processing and the 9th International Joint Conference on Natural Language Processing (EMNLP-IJCNLP), pages 5185–5194, Hong Kong, China. Association for Computational Linguistics.

[2] Zhaocheng Zhu, Zuobai Zhang, Louis-Pascal A. C. Xhonneux, Jian Tang: Neural Bellman-Ford Networks: A General Graph Neural Network Framework for Link Prediction. NeurIPS 2021: 29476-29490

[3] Sanxing Chen, Xiaodong Liu, Jianfeng Gao, Jian Jiao, Ruofei Zhang, and Yangfeng Ji. 2021. HittER: Hierarchical Transformers for Knowledge Graph Embeddings. In Proceedings of the 2021 Conference on Empirical Methods in Natural Language Processing, pages 10395–10407, Online and Punta Cana, Dominican Republic. Association for Computational Linguistics.

**Summary Of The Paper:**

This paper presents a thorough empirical investigation of the performance of knowledge graph embedding models on a variety of graph structure prediction tasks, including link prediction, relation prediction, domain prediction, and neighborhood prediction. The results show that these models do not perform well on the tasks except link prediction. The authors go on to propose and test multi-task learning to solve this problem. Multi-task learning can also help with KG downstream tasks like entity classification and regression.

**Summary Of The Review:**

Overall, this paper sheds some light on both KG embedding models and their applications. However, most of the findings, in my opinion, are common knowledge in the related field. Some recent representative KG embedding models and in-depth analysis of the reasons for the performance are lacking in the empirical study. So, I think the contribution of the paper is not enough.

---

> ### Author Response · Authors · 2022-11-18
> **Reply to your concerns**
>
> > Concern: It's not surprising that including MTT objectives improves on graph-structure prediction.
>
> Yes, we agree and, as you write, state this explicitly in the paper. Generally, our work does not aim to be surprising but to provide solid empirical evidence. Also note that we use graph-structure prediction for two purposes: (1) to explore the information captured in the KGE models and (2) to explore the impact model performance on downstream tasks (e.g., KGE models are used in industry in this way [1, 2]).
>
> > Concern: It's not surprising that KGE models trained on LP perform badly on graph-structure prediction, because there is a gap in the training and evaluation tasks.
>
> The gap is not as large as it may seem and at least we did not expect such low performance. The LP taks asks questions like (Austin, located in, ?) or (Austin, capital of, ?); we expect the model to answer Texas. The NBE task asks which entities Austin is related with; we also expect the model to answer Texas. This latter task is a seemingly simpler because the model does not need to distinguish relations, yet KGE models are bad at it. Similar arguments can be made for other tasks. E.g., a model may be able to answer (Austin, capital of, ?) and (?, capital of, Texas), yet it may not be able to predict that "capital of" is a relation connected to Austin and/or Texas (NBR).
>
> A main goal of our paper is to identify, quantify, and improve on these deficiencies in graph-structure prediction as well as their impact on downstream tasks.
>
> > Concern: Models are old.
>
> Yes. We nevertheless feel that these were the most suitable models for this study. Note that the oldest model (TransE 2013) fell behind new models (Complex 2016, RotatE 2019) on link prediction, but nevertheless often provided better performance on downstream tasks.
>
> We added some discussion on our choice of models to the experimental section. Quoting: "RotatE and ComplEx are the methods of choice for low-cost embeddings with good prediction performance [3], and---with an increase in model size and/or training cost [4,5]--- can perform as well as SOTA models of other KGE model types such as the transformer-based HittER model [6]."
>
> Some newer KGE models exclusively focus on link prediction; most notably, the HittER and NBFnet models that you cite. HittER's main idea is to add entity context for link prediction. It's not immediately clear how to use such a model for graph-structure prediction or downstream tasks. Likewise, NBFnet exclusively focus on link prediction and does not provide entity embeddings for downstream models. We instead considered KE-GCN as a suitable GNN baseline.
>
> > Concern: In-depth analysis is lacking.
>
> We've completely revised the discussion in the experimental section and added an experiment on task selection to provide further analysis and insight. We also greatly appreciate suggestions on how to further improve the discussion.
>
> [1] Ahmed El-Kishky, Thomas Markovich, Serim Park, Chetan Verma, Baekjin Kim, Ramy Eskander, Yury Malkov, Frank Portman, Sofı́a Samaniego, Ying Xiao, and Aria Haghighi. Twhin: Embedding the twitter heterogeneous information network for personalized recommendation. In Proceedings of the 28th ACM SIGKDD Conference on Knowledge Discovery and Data Mining, pp. 2842–2850, 2022.
>
> [2] Ihab F Ilyas, Theodoros Rekatsinas, Vishnu Konda, Jeffrey Pound, Xiaoguang Qi, and Mohamed Soliman. Saga: A platform for continuous construction and serving of knowledge at scale. 2022.
>
> [3] Daniel Ruffinelli, Samuel Broscheit, and Rainer Gemulla. You can teach an old dog new tricks! on training knowledge graph embeddings. In International Conference on Learning Representations, 2020
>
> [4] Timothée Lacroix, Nicolas Usunier, and Guillaume Obozinski. Canonical tensor decomposition for knowledge base completion. In International Conference on Machine Learning, pp. 2863–2872. PMLR, 2018.
>
> [5] Yihong Chen, Pasquale Minervini, Sebastian Riedel, and Pontus Stenetorp. Relation prediction as an auxiliary training objective for improving multi-relational graph representations. In 3rd Conference on Automated Knowledge Base Construction, 2021b.
>
> [6] Sanxing Chen, Xiaodong Liu, Jianfeng Gao, Jian Jiao, Ruofei Zhang, and Yangfeng Ji. Hitter: Hierarchical transformers for knowledge graph embeddings. In Proceedings of the 2021 Conference on Empirical Methods in Natural Language Processing, pp. 10395–10407, 2021a.

---

### Author Response · Authors · 2022-11-18
**We thank the reviewers for their work**

We thank all reviewers for their valuable comments and constructive criticism, which we found helpful to revise and improve the paper.

In addition to various clarifications and small changes, we revised and expanded the discussion in our experimental study and added new ablation results. All relevant changes are marked in blue (not including fixes to typos and minor writing issues).

We provide our thoughts on the individual points raised by each reviewer in separate responses.

---

### Author Response · Authors · 2022-11-30
**We'd greatly appreciate the reviewers' thoughts on our revised paper**

Dear reviewers,

we hope that our thoughts on your comments, the revised paper, as well as the new material added to it were helpful (expanded discussion in Section 4 and new ablation results). So far, we haven't heard back from you, neither via discussion nor via an updated review or scores. We'd greatly appreciate if you'd share your thoughts with us! We are also happy to provide further feedback and engage in discussions with you.

All the best,
The authors

---

### Decision · Program_Chairs · 2023-01-20

**Decision:**

Reject

**Justification For Why Not Higher Score:**

overall contributions are felt limited, unsurprising, so no reviewer championed the paper

**Justification For Why Not Lower Score:**

N/A

**Metareview: Summary, Strengths And Weaknesses:**

The paper makes one key observation that the KGE models trained only for link prediction are unsuitable for other related tasks. So, the paper proposes some other tasks and trains embeddings in multi-task framework, to show overall improvement in all tasks.

All reviewers appreciate the key insight. However, none of the reviewers feel that the paper has any wow effect. The downstream path from the observation of training multi-task is expected and obvious. And the results in many cases show marginal improvements. This reduces excitement on the paper.

There is also the added issue that the authors have very substantially revised the experimental section and added new experiments, etc. However, I feel that the additions to the original paper should be limited, and if one makes substantial changes then the revised version should go through another round of full review. In this paper's case, unfortunately, none of the reviewers responded to the improvements, and so we cannot be sure how to judge the revised paper.

In light of that, I suggest that the paper be rejected for this iteration. One, because the results are not entirely surprising, which limits the overall contribution of the work. And two, because the new changes probably need another round of review.

I also have some thoughts on 1-2 of the points raised by reviewers. Note that these are my individual thoughts, and should be used as just one other person's opinion (may helpful in preparing your next draft).
1) did you use a good set of models to start with? I'd personally say that the use of ComplEx and RotatE makes your models near state of the art. I checked the Tucker paper and noted that Complex's old results are used there and better implementations are not used. You may consider justifying your choice of models by the following two references (worth reading). Plus, you may limit your study to "tensor factorization" style models to avoid bringing some of the more recent models, like neural bellman ford and rnnlogic etc. into consideration.
- https://madoc.bib.uni-mannheim.de/54954/1/you_can_teach_an_old_dog_new_tricks_on_training_knowledge_graph_embeddings.pdf
- https://arxiv.org/abs/2005.00804

2) you say in your response that additional improvements may be possible with more study. I believe the onus is on you to convince the reviewers... maybe it makes sense to achieve more than marginal improvements, and increase the strength of the contributions before resubmitting the paper.